# Natural Adversarial Objects

**Felix Lau**
Scale AI
felix.lau@scale.com

Sasha Harrison
Scale AI
sasha.harrison@scale.com

Nishant Subramani
Intel Intelligent Systems Lab
nishant.subramani23@gmail.com

Aerin Kim
Scale AI
aerin.kim@scale.com

Elliot Branson
Scale AI
elliot.branson@scale.com

Rosanne Liu
ML Collective
rosanne@mlcollective.org

## Abstract

Although state-of-the-art object detection methods have shown compelling performance, models often are not robust to adversarial attacks and out-of-distribution data. We introduce a new dataset, Natural Adversarial Objects (NAO), to evaluate the robustness of object detection models. NAO contains 7,936 images and 13,604 objects that are unmodified, but cause state-of-the-art detection models to misclassify with high confidence. The mean average precision (mAP) of EfficientDet-D7 drops 68.3% when evaluated on NAO compared to the standard MSCOCO validation set. We investigate why examples in NAO are difficult to detect and classify. Experiments of shuffling image patches reveal that models are overly sensitive to local texture. Additionally, using integrated gradients and background replacement, we find that the detection model is reliant on pixel information within the bounding box, and insensitive to the background context when predicting class labels.

## 1 Introduction

It is no longer surprising to have machine learning vision models perform well on large scale training sets and also generalize on canonical test sets coming from the same distribution. However, generalization towards difficult, out-of-distribution samples still poses difficulty. Recht et al. [16] showed that model performance on canonical test sets is an overestimate of how they will perform on new data. Moreover, recent research on adversarial attacks has shown that deep neural networks are surprisingly vulnerable to artificially manipulated images, casting new doubt on the efficacy and security of such models.

The vulnerability of neural networks to adversarial attacks that are deliberately generated to fool the system is unsurprising, and well studied. However, this type of attack represents a narrow threat model because it necessitates that the adversary has control over the raw input, or has access to the model weights. It is often overlooked that real-world, unmodified images can also be used adversarially to cause models to fail. These "natural" adversarial attacks represent a less restricted threat model, where an attacker can easily create black-box attacks without carefully constructing input perturbations [5], but only by using naturally occurring images that are easily obtainable. Such images are called natural adversarial examples [7]: unmodified, real-world images that cause modern image classification models to make egregious, high-confidence errors.

In [7] natural adversarial examples are only constructed for image classification models. In this work, we seek to create an evaluation set analogous to [7], but instead targeted at object detection tasks. We name such a dataset Natural Adversarial Objects (NAO). The goal of NAO is to benchmark the worst case performance of state-of-the-art object detection models, while requiring that examples included in the benchmark are unmodified and naturally occurring in the real world.

Submitted to the 35th Conference on Neural Information Processing Systems (NeurIPS 2021) Track on Datasets and Benchmarks. Do not distribute.

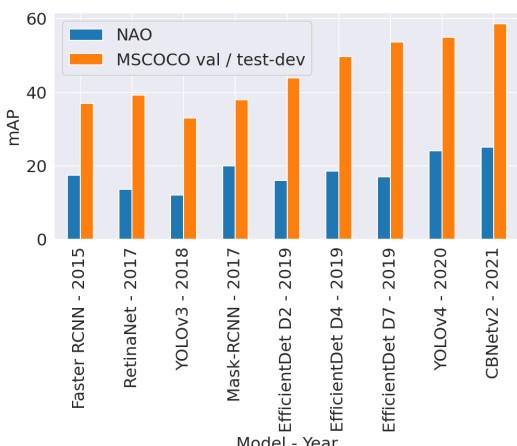

Figure 1: Mean average precision (mAP) of various detection models evaluated on NAO and MSCOCO *val* or *test-dev* set. All models show significant reduction in performance on NAO despite their accuracy improvement in MSCOCO in recent years. NAO is a challenging test set for detection models trained on MSCOCO and future work is required to close the performance gap.

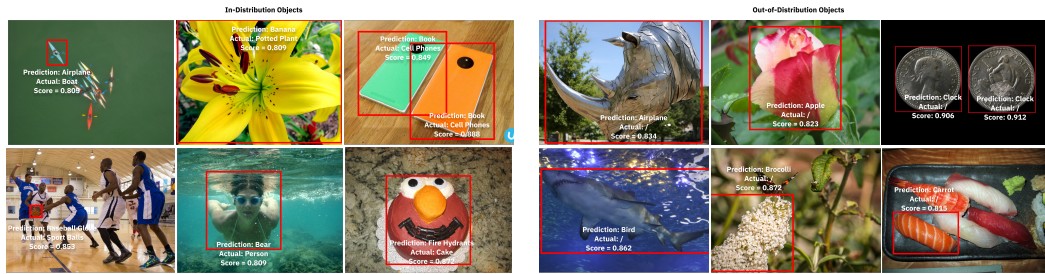

Figure 2: Sample images from NAO where EfficientDet-D7 produces high confidence false positives and egregious classification. **Left:** High confidence misclassified objects where the ground truth label is in-distribution and among the MSCOCO object categories. **Right:** High confidence false positives where the ground truth object is out-of-distribution (i.e. not part of MSCOCO object categories). The misclassified objects and false positives are superficially similar to the predicted classes – for example, the fin of the shark is visually similar to the airplane tail and the yellow petals of the flower is similar to a bunch of bananas.

We present a method to identify natural adversarial objects using a combination of existing object detection models and human annotators. First, we compare the predictions from various off-the-shelf detection models against a dataset already annotated with ground truth bounding boxes. We consider images containing high confidence false positives and misclassified objects as candidates for NAO. Then, we use a human annotation pipeline to filter out mislabeled images and non-obvious objects (e.g. occluded or blurry objects). Finally, we re-annotate the images using the object categories of the Microsoft Common Objects in Context (MSCOCO) dataset [13].

We perform extensive analyses to understand why objects in NAO are naturally adversarial. We visualize the embedding space common to MSCOCO, OpenImages, and NAO, and show that NAO images exist in the "blind spots" of the MSCOCO dataset. Next, by comparing integrated gradients [23] with predicted bounding boxes and replacing object backgrounds, we show that the detection model seldom makes use of object contexts. Lastly, by shuffling patches within the bounding box, we show that models relies on object subparts and texture to detect and classify the objects.

NAO ground truth bounding box annotations are made available under CC-BY 4.0.

## 2 Related Work

**Natural Adversarial Examples**   Hendrycks et al. [7] construct two datasets, namely *ImageNet-A* and *ImageNet-O*, to measure the robustness of image classifiers against out of distribution examples. To construct these two datasets, they choose images on which a pretrained ResNet model failed to make a correct prediction. We adopt a similar approach for selecting adversarial examples but use an object detection model and take extra steps to ensure high quality annotations by using human annotators. Zhao et al. [27] develops a method to generate adversarial perturbation that lies on the data manifold where the pertubation is meaningful to the semantic of the images.

**Adversarial Examples**   Adversarial examples are inputs that are specifically designed to cause the target model to produce erroneous outputs. Arpit et al. [1] analyzed the capacity of neural networks to memorize training data, and found that models with a high degree of memorization are more vulnerable to adversarial examples. Jo and Bengio [9] have shown that convolutional neural networks tend to learn the statistical regularities in the training dataset, rather than the high level abstract concepts. Since adversarial examples are transferable between models that are trained on the same dataset, these different models may have learned the same statistics and therefore are vulnerable to similar adversarial attacks. Brendel and Bethge [3] show that small local image features are sufficient for deep learning model to achieve high accuracy. Geirhos et al. [4] show that ImageNet-trained CNNs are biased toward texture and created *Stylized-ImageNet* to reveal the severity of such bias. Similarly, Ilyas et al. [8] showed that adversarial examples are a byproduct of exploiting non-robust features that exist in a dataset. Non-robust features are derived from patterns in the data distribution that are highly predictive, yet brittle and incomprehensible to humans. Undoubtedly, the reasons behind the existence and pervasiveness of adversarial examples still remains an open research problem. Zhang and Wang [26] developed a method to improve the robustness of object detection model by identifying an asymmetric role of task losses.

**Model Interpretability**   While the interpretability of deep neural networks remains an open research question, there exist attribution methods that help explain the relationships between the input and output of such models. Sundararajan et al. [23] suggests that attribution methods should satisfy two axioms: sensitivity and implementation invariance, and proposes a new method, *Integrated Gradient*, to understand which parts of an image influence the prediction the most.

**Object Detection Architectures**   Detection models fall into two categories: one-stage ([25], [18], [17]) and two-stage models ([20], [22]), differentiated by whether the model has a region pooling stage. Single-stage model are more computationally efficient, but usually less accurate than the 2-stage models. In this paper, we evaluate both single and two-stage models using the NAO dataset. Tan et al. [25] introduced EfficientDet, which uses EfficientNet [24] as backbone and uses BiFPN such that the model is more efficient while more accurate, achieving state-of-the-art results in MSCOCO at 54.4 on the *val* set.

## 3 Creating Natural Adversarial Objects (NAO) Dataset

### 3.1 Limitations of MSCOCO

MSCOCO [13] is a common benchmark dataset for object detection models. It contains 118,287 images in the training set, 5,000 images in the *val* set and 20,288 images in the *test-dev* set. MSCOCO contains 80 object categories consisting of common objects such as *horse*, *clock*, and *car*. The goal of MSCOCO is to introduce a large-scale dataset that contains objects in non-iconic or non-canonical views. The images in MSCOCO were originally sourced from Flickr, then filtered down in order to limit the scope of the benchmark to a set of 80 categories. These 80 categories were chosen from a list of the most commonplace visually identifiable objects. Still, this category list represents only a small subset of object categories in real life. For example, 'fish' is not among the 80 categories, and as a result there are only a few photos taken underwater. This leads to a biased benchmark with limitations for generalizability and robustness. As a result, in this work, we ensure more diverse sourcing — choosing images from OpenImages v6 [11], a dataset with 600 object categories, in order to create a more representative dataset.

| | Statistics | | Top 3 Objects (Count) | | |
|---|---|---|---|---|---|
| | # of Images | # of Objects | 1st | 2nd | 3rd |
| MSCOCO *val* | 5,000 | 36,781 | Person (11,004) | Car (1,932) | Chair (1,791) |
| MSCOCO *test-dev* | 20,288 | - | - | - | - |
| NAO | 7,936 | 13,604 | Person (4,693) | Cup (2,257) | Car (752) |

Table 1: Dataset statistics of MSCOCO *val*, *test-dev* and NAO.

## 3.2 Sourcing Images for NAO from OpenImages

To create NAO, we first sourced images from the training set of OpenImages [11], a large, annotated image dataset containing approximately 1.9 million images and 15.8 million bounding boxes across 600 object classes.

One challenge of using OpenImages is that the bounding boxes are not exhaustively annotated. Each image is first annotated with positive and negative labels which indicate the presence or absence of an object in the image. Only objects belonging to the positive label categories are annotated with bounding boxes. As a result, some objects that belong to the OpenImages object categories are not labeled with a bounding box. For example, imagine both horse and pig are represented in the 600 object classes. If an image contains a horse and a pig, and only the category of horse is included in preliminary round of positive labels, then the image would be labeled with a bounding box for the horse but not the pig. This non-exhaustive annotation approach makes it difficult to produce and compare precision and recall to other exhaustively annotated dataset such as MSCOCO. This is because false positives and false negatives can only be evaluated accurately if the ground truth bounding boxes are exhaustive.

One other challenge that arises when sourcing images from OpenImages is that the object categories of OpenImages and MSCOCO are not the same. Therefore, after obtaining a set of natural adversarial images, we exhaustively annotate the images with all 80 MSCOCO object classes to facilitate straightforward comparison between NAO and the MSCOCO *val* and *test* sets.

## 3.3 Candidate Generation

To generate object candidates, we perform inference on OpenImages using an EfficientDet-D7 model [25] pretrained on MSCOCO, which yields predicted object bounding boxes for each candidate image. Our goal is to find two types of errors: (i) **hard false positives** (i.e. false positives with high confidence) and (ii) **egregiously misclassified** objects. For a detection to be a hard false positive, we require the prediction to have no matching ground truth box with intersection over union (IoU) greater than 0.5, and class confidence score greater than 0.8. We define egregiously misclassified objects as predictions that have a matching ground truth bounding box with an IoU greater than 0.5, but have an incorrect classification with a confidence greater than 0.8. We do not consider false negatives with high confidence because we observe that these are commonplace especially in crowded scenes. There are 43,860 images containing at least one hard false positive or egregiously misclassified object.

## 3.4 Annotation Process

Our annotation process has two annotation stages: classification and bounding box annotation.

**Classification Stage**  In the classification stage, annotators identify whether the object described by the bounding box shown indeed belongs to the ground truth class as defined by the annotation in OpenImages or as predicted by the EfficientDet-D7. The purpose of this stage is to remove the possibility that the model prediction is "incorrect" due to the ground truth label being incorrect.

In addition, we ask the annotators to confirm whether the object can be "obviously classified" according to the following criteria:

1. Is the bounding box around the object correctly sized and positioned such that it is not too big or too small?

2. Does the object appear blurry?

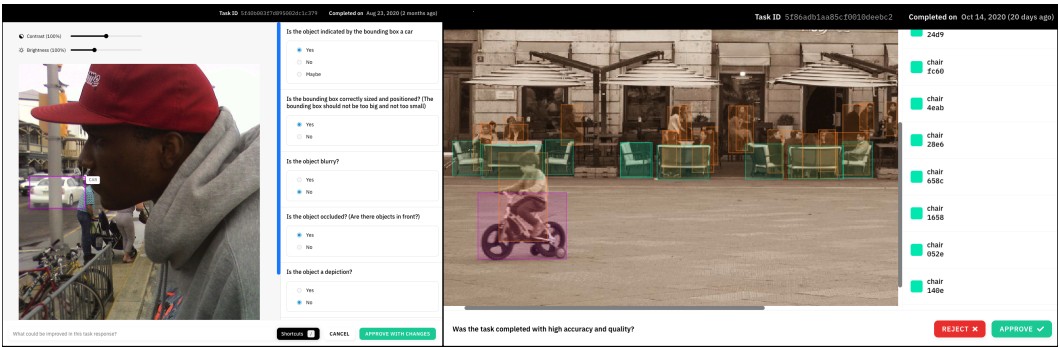

Figure 3: **Left**: Annotation interface for the first annotation stage (classification) where the annotator confirms that the object belongs to the correct category, not occluded, not blurry and not a depiction. **Right**: Annotation interface for second annotation stage (bounding box) where the annotators locate and classify all objects in the images using the MSCOCO object categories.

3. Is the object occluded (i.e. are there other objects in front of this one)?

4. Is the object a depiction of the correct class (such as a drawing or an image on a billboard)?

We ask these additional questions to filter out ambiguous objects, such that a human can easily identify what class an object belongs to. After this filtering, 18.1% of the images (7,936) remain; each of the remaining images are confirmed to fulfill the 4 criteria, and represent true misclassifications by the model. In this first annotation stage (classification), 5 different annotators are asked to annotate the same image and we use their consensus to produce an aggregated response by majority vote.

**Bounding Box Stage**  In the second annotation stage (bounding box), annotators exhaustively identify and put boxes around all objects that belong to the MSCOCO object categories. We are unable to directly use the annotations from OpenImages because there is not a one-to-one mapping between the OpenImages and MSCOCO object categories, and because the bounding box annotations from OpenImages are not exhaustively annotated. However, the bounding box annotations from OpenImages are provided to the annotators as a starting point.

These bounding box annotation tasks are completed by 2 sets of annotators. The first set of annotators complete the bulk of the task by placing bounding boxes around objects that belong to the MSCOCO object categories. The second set of annotators review the work of the first set of annotators, sometimes adding missing bounding boxes or editing the existing ones.

To ensure the quality of the annotation is high, in both of these stages, the annotators have to pass multiple quizzes before they can start working tasks to ensure they understand the instructions well. If the annotator fails to maintain a good score, they are no longer eligible to continue to annotate the images. This process of vetting annotators is consistent with the methodology used to construct MSCOCO [13].

When the annotators from the 2 different stages disagree, we tie break by choosing second annotator who is positioned as the reviewer.

### 3.5 Evaluation Protocol

The goal of NAO is to test the robustness of object detection models against edge cases and out-of-distribution images. We propose two main evaluation metrics: **overall mAP** and **mAP without out-of-distribution objects**. mAP without out-of-distribution objects evaluates against edge cases of object categories that the detection models are trained on, while the overall mAP evaluates robustness against out-of-distribution objects. For calculating mAP without out-of-distribution objects, any detection matched to an object not belong to the 80 MSCOCO object categories is not considered a false positive.

NAO should be mainly used as a test set to evaluate detection models trained on MSCOCO. However, a split of train, validation and test set is also provided for robustness approaches that require training.

| | | MSCOCO *val* | MSCOCO *test-dev* | NAO | | |
|---|---|---|---|---|---|---|
| | Params | mAP | mAP | mAP | mAR | mAP w/o OOD |
| Faster RCNN [19] | 42M | 21.2 | 21.5 | 17.4 | 48.9 | 28.2 |
| RetinaNet-R50 [14] | 34M | 39.2 | 39.2 | 13.7 | 41.1 | 22.8 |
| YOLOv3 [18] | 62M | - | 33.0 | 12.1 | 30.4 | 19.6 |
| Mask RCNN R50 [6] | 44M | 37.9 | - | 20.0 | 51.4 | 30.6 |
| EfficientDet-D2 [25] | 8.1M | 43.5 | 43.9 | 16.1 | 46.1 | 28.6 |
| EfficientDet-D4 [25] | 21M | 49.3 | 49.7 | 18.6 | 50.0 | 34.3 |
| EfficientDet-D7 [25] | 52M | 53.4 | 53.7 | 17.0 | 46.3 | 30.6 |
| YOLOv4-P7 [2] | 28.7M | 55.3 | 55.5 | 24.1 | 62.0 | 41.6 |
| CBNetv2-HTC [12] | 231M | 58.2 | 58.6 | 25.1 | 61.5 | 43.2 |

Table 2: mAP of various detection models evaluated on MSCOCO *val* and *test-dev* set and NAO. Accuracy of all models were significantly lower on NAO than on MSCOCO. There is a slight increase in mAP when out-of-distribution (OOD) objects are excluded.

## 4 Results

### 4.1 Evaluation of Detection Models

Figure 1 and Table 2 show the mean average precision (mAP) of several state-of-the-art detection models evaluated on MSCOCO and NAO. Despite the fact that the images in NAO were chosen using an EfficientDet-D7 model, we observe that other object-detection architectures show a similar reduction in mAP when evaluated on NAO. Concretely, when using NAO the mAP of EfficientDet-D7 is reduced by 68.3%, while Faster RCNN is reduced by 19.1% when compared to MSCOCO. Even though EfficientDet-D7 was developed more recently than Faster RCNN, the mAP on NAO is similar. This indicates that latest models are not more robust on NAO, despite their superior performance on MSCOCO evaluation sets. This in turn suggests that modeling improvements from recent years do not address the issue of high confidence misclassification in out-of-distribution samples.

We also calculate the mAP without out-of-distribution objects. That is, if a detection matches a bounding box that does not belong the MSCOCO object categories, the detection is not counted as a false positive. We can see that, this exclusion improves mAPs on NAO, but overall, the results are still considerably worse than those from the MSCOCO *val* and *test-dev* set.

### 4.2 Common Failure Modes

In Figure 4, we visualize some failure modes of the detection models on NAO. In most of the misclassified objects, the predicted class is superficially similar to the ground truth class, but obviously different in terms of function. For example, clocks and coins are similar in shape (circular), texture (metallic in some cases) and both have characters near the perimeters. However, they are very different in function and in scale, such that any human can easily tell the difference between the two. Similarly, airplanes and sharks are similar in overall shape, color, and texture, but exist in rather different scenes.

Another common failure mode is differentiating different animal species. For example, elephant and rhinoceros both have somewhat similar skin color and texture but they are very different in size and rhinoceros do not have the distinctive elephant trunk.

### 4.3 Dataset Blind Spot

As mentioned in Section 3.1, MSCOCO sourced images from Flickr search queries related to the 80 object categories. This process can be seen as a biased sampling process of all captured photos, resulting in "blind spots" in MSCOCO. For example, because there is not any "fish" category, the frequency of photos taken underwater in MSCOCO is much lower than all captured photos. In this section, we investigate this sampling bias by comparing the image embeddings of BiT ResNet-50 [10] pretrained on ImageNet-21k [21] across the 3 datasets – OpenImages *train*, MSCOCO *train* and NAO. We consider OpenImages as a universal set for available images and with MSCOCO and NAO being a subset of the captured images. The image embedding is the output of the global average

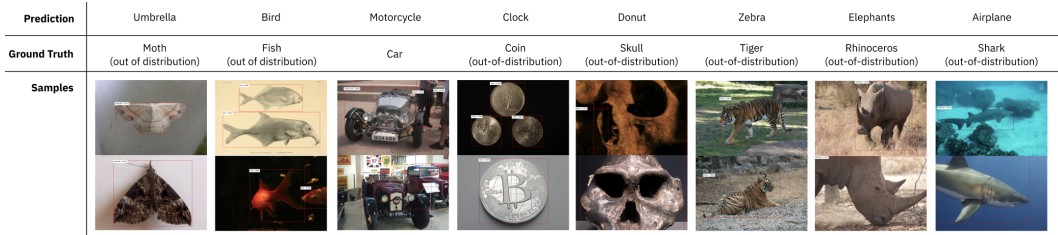

| Prediction | Umbrella | Bird | Motorcycle | Clock | Donut | Zebra | Elephants | Airplane |
|---|---|---|---|---|---|---|---|---|
| Ground Truth | Moth (out of distribution) | Fish (out of distribution) | Car | Coin (out-of-distribution) | Skull (out-of-distribution) | Tiger (out-of-distribution) | Rhinoceros (out-of-distribution) | Shark (out-of-distribution) |
| Samples | | | | | | | | |

Figure 4: Selected samples to showcase common failure modes.

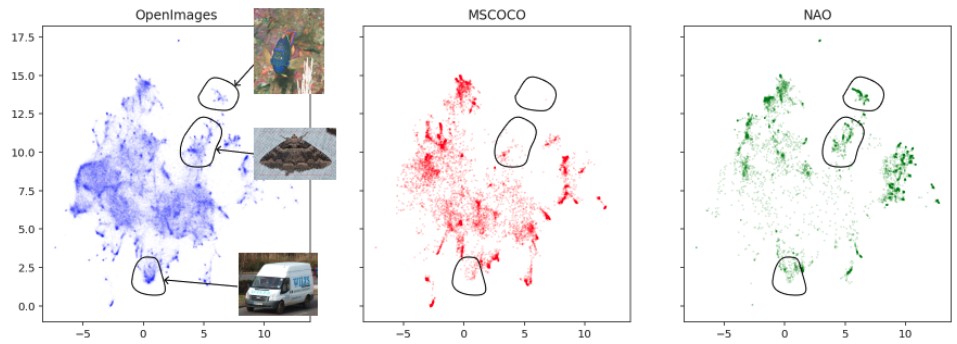

Figure 5: BiT ResNet-50 embeddings projected by UMAP on OpenImages *train*, MSCOCO *train* and NAO. NAO images are under-represented in MSCOCO.

pooling layer, resulting in a vector of size 2,048. We then use UMAP [15] to reduce the dimension to 2 for visualization as shown in Figure 5.

When comparing the embedding space of MSCOCO with OpenImages, we found that there are regions where the density is significantly lower in MSCOCO than in OpenImages. Some of these low-density regions are indicated by the black circles in Figure 5. When cross-referencing with the embedding space of NAO, we can see that these low-density regions of MSCOCO are in fact high-density in NAO, indicating that the examples in NAO are exploiting the under-represented regions that arise from MSCOCO's biased sampling process. We visualize 3 of such low-density clusters and they each reveal a common failure mode (i.e. fish misclassified as bird, insects misclassified as umbrella and van misclassified as truck.)

## 4.4 Background Cues

Hendrycks et al. [7] suggest that classification models are vulnerable to natural adversarial examples because classifiers are trained to associate the entire image with an object class, resulting in frequently appearing background elements being associated with a class. Object detection models are different from image classifiers in that they receive additional supervision about the object position and size. We instead argue that the primary cause of detection models being vulnerable to NAO is their tendency to focus too much on the information within the predicted bounding boxes.

In this section, we study the effect of object background on classification probabilities. Specifically, we quantify the change in probability of the detected object when its background is replaced. We use a MSCOCO-pretrained Mask-RCNN [6] with a ResNet 50 backbone to obtain instance segmentation masks on MSCOCO *val* and NAO. Then, we use the instance segmentation masks to retain only the most confident object and replace the rest of the image with a new background. There are 6 new backgrounds – underwater, beach, forest, road, mountain and sky – where Mask-RCNN detects no objects of probability higher than 0.1 from the backgrounds themselves. We measure the change of probability by matching the bounding box detected on the original image and the bounding box

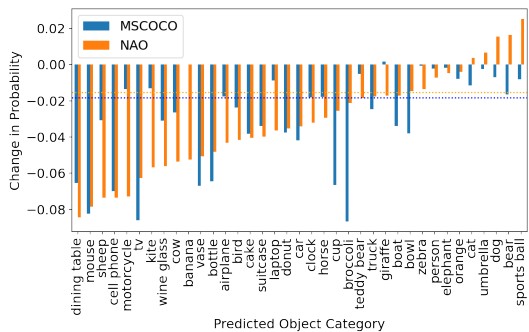

Figure 6: Average change in probability of objects when the backgrounds are replaced. The orange and blue dotted lines indicate average change in probabilities across all classes in MSCOCO and NAO. The small change in probability indicates that the detection model did not make use of background to classify the objects.

detected on the new image with the background replaced. We repeat this process for all images in NAO and MSCOCO *val* set and all 6 backgrounds.

As show in Figure 6, in both NAO and MSCOCO, the change in probabilities is low, indicating that the model does not make use of the background when detecting the object. While this robustness against background change is favorable in most cases, this also shows that the model also does not account for unlikely combinations of background and foreground objects. For example, when the model misclassifies a shark as an airplane, the network could have noticed that the detected "airplane" is underwater and assigned a lower probability to the class airplane.

## 4.5 Integrated Gradients Analysis

We further try to understand the source of the egregious misclassifications by computing the integrated gradients [23] of the network classification head output with respect to the input image. We aim to find the proportion of integrated output within the bounding box to understand if the network makes use the context of the object for detection and classification.

Specifically, we computed the gradients of the classification output of highest-scored bounding box with respect to the input image and measure the proportion of the sum of attribution inside the bounding box with respect to the total attribution. When there are multiple same-class objects to detect, we make sure to attribute each object separately. For example, when there are 2 people, we calculate the attribution of one person, ensuring the attribution of the other person is not counted towards the background. We used EfficientDet-D4 and randomly sampled 1000 images for this experiment. We found that for most classes, the majority of the attributions come from inside the bounding box.

Both Figure 6 and Figure 7 suggest that the detection model do not make use of background enough and instead mainly focus on the information within the predicted bounding box.

## 4.6 Patch Shuffling and Local Texture Bias

Geirhos et al. [4] demonstrated that classification networks are biased towards recognizing texture instead of shape. Brendel and Bethge [3] showed that a classification network can still reach a high level of accuracy using just small patches extracted from images. In this work, we show that detection models also show strong local texture bias, making them susceptible to adversarial objects with similar object subparts but are of another object category. For each prediction from EfficientDet-D7 on MSCOCO and NAO, we randomly sample a patch from within the bounding box and swap it with another patch from inside the bounding box. We swap these random patches 3 times such that object is barely recognizable by just the shape. We then match the detected bounding box from the shuffled images to the original bounding box with the highest overlap. We repeat this shuffling process independently 10 times and record the absolute change in probability of the detected object. Figure 8 shows that there is only a modest reduction in probability after the shuffles.

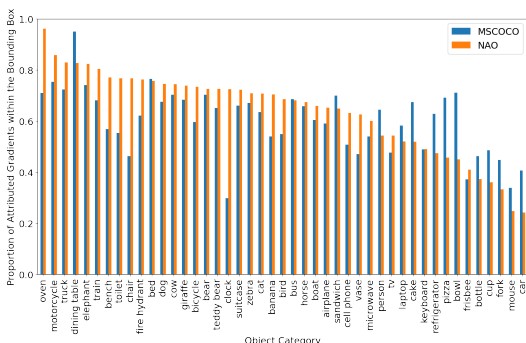

Figure 7: Proportion of attributed gradients within the bounding box by object category. In many classes, the detection model seldom make use of the object surroundings for classification and detection.

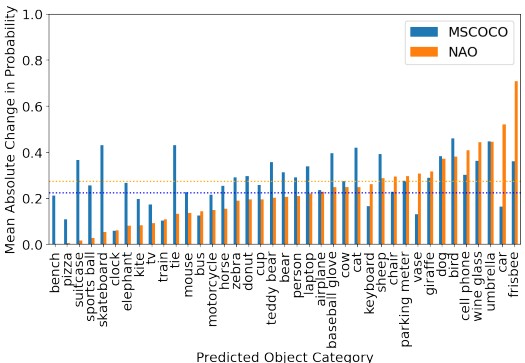

Figure 8: Mean absolute change in probability when patches inside the bounding box are swapped randomly. The blue and orange dotted line represent the mean average change in probability across all object categories for MSCOCO and NAO respectively. This confirms the texture bias hypothesis because even if the shape of the objects are heavily distorted while the local texture is intact, the network is still able to make the same prediction with similar confidence in most object categories.

## 5    Conclusion

We introduce "Natural Adversarial Objects" (NAO), a challenging robustness evaluation dataset for detection models trained on MSCOCO. We evaluated seven state-of-the-art detection models from various families, and show that they consistently fail to perform accurately on NAO, comparing to MSCOCO *val* and *test-dev* set, including on both in-distribution and out-of-distribution objects. We explained the procedure of creating such a dataset which can be useful for creating similar datasets in the future.

We expose that these naturally adversarial objects are difficult to classify correctly due to the "blind-spots" in the MSCOCO dataset. We also utilize integrated gradients, background replacement, and patch shuffling to demonstrate that detection models are overly sensitive to local texture but insensitive to background change, leading such models to be susceptible to natural adversarial objects. We hope NAO can facilitate further research about model robustness and handling out-of-distribution inputs.

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
