# OpenReview forum: "Natural Adversarial Objects"
_NeurIPS.cc/2021/Track/Datasets_and_Benchmarks/Round1 — Submitted to NeurIPS 2021 Datasets and Benchmarks Track (Round 1)_

### Official Review · Reviewer_XeSN · 2021-06-28
**The authors present a new dataset to evaluate the robustness of object detection models. The dataset is OK but the detailed description for the use and maintenance plan is missing.**

**Rating:** 6
**Confidence:** 4
**Correctness:** Seems correct.
**Clarity:** The paper is well written.

**Strengths:**

 -Filtering and Correcting the low-quality labels in existing datasets (e.g., MSCOCO) is meaningful however time-consuming work.

 -It is good to establish an annotation interface for the annotators.

**Weaknesses:**

-The scale of the proposed NAO is relatively small, compared to MSCOCO, OpenImage, etc.

-The benchmark models are outdated. More models should be included, e.g., YOLOv4, DyHead, etc. Please refer to https://paperswithcode.com/sota/object-detection-on-coco for details.

-Low-quality figures. The text is hardly seen in some of the figures, e.g., fig.2, fig.3, fig.4. Please select the appropriate font size.

-The document of the dataset is not enough.

**Additional Feedback:**

Refer to the Weaknesses section.

**Documentation:**

There is no detailed readme file in the provided URL. The licensing and maintenance plan is also not provided.

**Ethics:**

No.

**Relation To Prior Work:**

Yes, it is.

**Summary And Contributions:**

The authors present a new dataset to evaluate the robustness of object detection models. The proposed dataset focuses on high false-positive and misclassified cases of MSCOCO, provides further manual annotations and benchmark studies to illustrate the insights.

---

> ### Author Response · Authors · 2021-07-09
> **Details (e.g. maintenance plan and description) are in Appendix**
>
> We thank reviewer XeSN for the review. Below are our responses:
>
> > The scale of the proposed NAO is relatively small
>
> NAO is sourced from 1.9 million images and the resulting filtered dataset is comparable to the validation set of MSCOCO. Also NAO is designed to be used mainly as a test set.
>
> > The benchmark models are outdated
>
> We have added 2 new models -- YOLOv4 and CBNetv2. We were not able to find an official implementation of DyHead as of the time of writing this response.
>
> > Low-quality figures
>
> We have updated our figures with larger font size and higher quality.
>
> > The document of the dataset is not enough
>
> The Appendix includes "Datasheet for Dataset" which contains lots of information about the construction and the composition of the dataset. We have added the datasheet to the Google Drive folder.
>
> > The licensing and maintenance plan is also not provided.
>
> Maintenance plan and licensing are stated in the Appendix. It will be maintained by the machine learning team at Scale AI and the license is CC-BY 4.0.

---

> > ### Comment · Reviewer_XeSN · 2021-07-20
> > **Response to authors' rebuttal**
> >
> > Thanks for the authors' response. After reading the responses, I decided to maintain the original rating as marginally above acceptance threshold.

---

### Official Review · Reviewer_Kcpe · 2021-07-01
**Interesting dataset, but the paper has several issues**

**Rating:** 5
**Confidence:** 3

**Strengths:**

The existence of adversarial examples in machine learning is a phenomenon with far-reaching consequences for, e.g., network security. Most of the research on adversarial examples has thus far been done on synthetic examples. Research on naturally occuring adversarial examples, while being equally (or even more) important, has thus far received much less attention. As a consequence, datasets geared to this specific issue are of tremendous importance. The dataset introduced by this paper could benefit and accelerate the research in this field.

The authors' analysis of possible reasons for the bad detection performance on the dataset can be a basis for further research on the nature of adversarial examples.

**Weaknesses:**

All evaluated detection models were pretrained on MSCOCO. It is not clear whether the objects in NAO would be adversarial for models trained on other detection datasets containing similar classes. Further analysis in this direction would improve the paper significantly. *Edited to add:* In their reply to this review, the authors pointed out the important role of MSCOCO in the field of object detection. I still think it would be interesting to analyse models trained on other datasets, too.

~~The discussion of the "dataset blind spot" (areas in the feature space with high density of OpenImages/NAO samples and low density of MSCOCO samples) in section 4.3 does not yield much insight into the question of adversarial objects. From the examples provided, it seems that NAO samples in these blind spots are from classes not present in MSCOCO. The fact that there are OOD samples in areas with few MSCOCO samples is to be expected and is not a very notable finding in itself.~~ *Edited to add:* In their reply to this review, the authors explained that the point of that section is not that there are OOD samples in the blind spots. It is that these blind spots contain OOD samples for which the model confidently predicts false positive bounding boxes. This is a valid, non-trivial observation.

**Additional Feedback:**

In line 205, the authors write "We consider OpenImages as a proxy for all captured images". This is a bold statement that should have some form of justification. *Edited to add:* The authors changed this sentence. However, I am still missing some explanation why the authors consider OpenImages as a universal set.

Typos:
in the Supplementary material, line 444: "will be" -> "were"
in the Supplementary material, line 569: "The images is" -> "The images are"

**Clarity:**

The paper is well-written and easy to follow. The figures and tables make sense and complete the text.

**Correctness:**

The mode of dataset construction is well-motivated and appropriate. Detection performance was measured with standard metrics. As a whole, the experiments are reasonably designed. There are a few issues as mentioned below:

The calculation of the mAP without OOD samples appears to have a problem. The authors explain that if a detection matches a bounding box that does not belong to the MSCOCO categories, the detection is not counted as a false positive. However, the images can still contain plenty of objects for which no annotated bounding boxes exist and which are out-of-distribution with respect to MSCOCO. As a consequence, the calculation of the "mAP without OOD" might still include lots of predictions based on OOD features.

The set of detection models evaluated on NAO includes EfficientDet D7, which was used in the generation of NAO. Low performance of EfficientDet D7 on NAO is to be expected, as the samples for NAO were selected specifically because EfficientDet D7 performed badly on them. ~~Therefore, EfficientDet D7 should not be part of the final evaluation. (An exclusion of EfficientDet D7 from the evaluation would not change the findings anyway, since all evaluated models perform badly on NAO).~~ *Edited to add:* As pointed out by the authors in their reply, the result on EfficientDet D7 has value as a reference. However, the role of EfficientDet D7 in the dataset creation should be clearly indicated, e.g., in Figure 1, so that a reader does not miss this fact when interpreting the results.

In Section 4.4, the authors describe that they use an MSCOCO-pretrained Mask-RCNN to obtain instance segmentation masks on MSCOCO and NAO. This is problematic since the evaluation of the paper shows that the Mask-RCNN performs badly on NAO. As a consequence, it is not clear whether the background replacement works correctly. *Edited to add:* In their reply, the authors argue why they think that this is not a big problem. While I still think that there is a problem, I do not think that the problem completely invalidates the authors' analysis in Section 4.4.

**Documentation:**

The dataset generation process is explained in sufficient detail.

Not all experiments can be reproduced with the information given in the paper alone. For example, the patch shuffling experiment in Section 4.6 cannot be reproduced without further information on the chosen parameters such as patch size etc.
However, the authors announce that the code will be released at a later date. This should make reproduction of the experiments possible (if code for all experiments is included).

Licensing (CC-BY), hosting and maintenance information is provided in the supplementary material. A link to the dataset has been included.

The supplementary material contains detailed information about the dataset. However, there seem to be some inconsistencies and missing information that could be fixed/completed.
Examples:
- In line 463, the authors write that "it is possible that some people [are] in frame of the images". This vague statement is strange considering that the object category "person" is the category with the most instances in the dataset.
- In line 507, the authors write that ethical review processes were conducted. An explanation of what kind of reviews mere made is missing.
- In line 541, the question "Is the software used to preprocess/clean/label the instances available?" is answered as "Not applicable". As software was used for labelling (as evidenced by the screenshots in figure 3), it is unclear why the question would not be applicable. If the software cannot be made available, the answer should be "No".

The authors might want to take another look at the information given in the supplementary material and extend/correct it where appropriate.

*Edited to add:* The authors updated the documentation and addressed the examples listed above.

**Ethics:**

No dedicated ethics review appears to be necessary.

**Relation To Prior Work:**

The authors discuss some related work regarding adversarial examples, model interpretability, and object detection architectures.

The paragraphs on (natural) adversarial examples are rather short and contain only few references. There is no discussion of previous work on adversarial examples for object detection (e.g., the following paper and references therein: Haichao Zhang, Jianyu Wang: "Towards Adversarially Robust Object Detection", ICCV 2019, Link: https://openaccess.thecvf.com/content_ICCV_2019/html/Zhang_Towards_Adversarially_Robust_Object_Detection_ICCV_2019_paper.html).

The authors might also consider mentioning some previous work on adversarial sample generation, e.g., Zhenglie Zhao et al.: "Generating Natural Adversarial Examples", ICLR 2018 (Link: https://openreview.net/forum?id=H1BLjgZCb).

*Edited to add:* The authors added some references. When citing the paper "Generating Natural Adversarial Examples" by Zhao et al., it might be good to mention that the "more natural" adversarial examples proposed by Zhao et al. are not "natural" in the sense of the paper under review.

**Summary And Contributions:**

This paper introduces a new objection detection image dataset named "Natural Adversarial Objects" (NAO). It contains images with objects that a state-of-the-art objection detection model trained on MSCOCO had difficulty to detect. The authors consider these objects as "naturally adversarial samples" (in contrast to often studied synthetically generated adversarial samples). These samples are sourced from the OpenImages dataset. A sample was selected if the object detection model generated a highly confident false positive or a misclassification.

An evaluation of a range of object detection networks on NAO shows a significant drop in performance compared to an evaluation on the MSCOCO dataset. A modified evaluation that tries not to count predicted bounding boxes on out-of-distribution objects as false positives yields a similar result.

Apart from the performance evaluation, the paper includes additional experiments that try to shine a light on what effects may be responsible for the bad detection performance on NAO.
By comparing feature embeddings of OpenImages, MSCOCO, and NAO images, the authors find that a lot of NAO images are in regions were there are few MSCOCO images.
Another experiment concerns the effect of object background on object classification. The authors find that replacing the background of objects changes classification probabilities only marginally and conclude that the model does not make much use of the background for object detection. The results of an integrated gradient analysis support this thesis.
Lastly, the authors study the importance of texture for the object classification by shuffling image patches inside the object bounding boxes. As this does not change classification probabilities by much, the authors infer that the detection models exhibit a strong bias towards texture, making the models susceptible to adversarial objects with similar texture.

---

> ### Author Response · Authors · 2021-07-09
> **Response to the issues raised**
>
> We thank reviewer Kcpe for the detailed and thoughtful comments. Our response are as follows:
>
> > All evaluated detection models were pretrained on MSCOCO.
>
> Despite being a relatively smaller dataset, MSCOCO is one of the most popular dataset to be evaluated against modern object detection networks. By evaluating models trained on MSCOCO, we reveal the fact that having improvement in accuracy on MSCOCO does not translate into model robustness. We hope our dataset can be used as a secondary test dataset to measure the robustness against natural adversarial objects and out-of-distribution objects.
>
> > The discussion of the "dataset blind spot" (areas in the feature space with high density of OpenImages/NAO samples and low density of MSCOCO samples) in section 4.3 does not yield much insight into the question of adversarial objects.
>
> The main takeaway of section 4.3 is that each blind spot (out of distribution objects) is indeed a common failure mode with high confidence. The fact that the model produce high confidence error in these blind spots is surprising.
>
> > The calculation of the mAP without OOD samples appears to have a problem.
>
> The meaning of OOD in this metric is objects that do not belong to the MSCOCO label categories. For example, if the model predicts a shark as an airplane, we do not penalize the model for this prediction because it has not seen a shark before. However, we recognize that if the airplane is OOD (e.g. an old propeller plane instead of a jet plane), this metric does not exclude such OOD objects that are part of the MSCOCO label categories.
>
> > In Section 4.4, the authors describe that they use an MSCOCO-pretrained Mask-RCNN to obtain instance segmentation masks on MSCOCO and NAO. This is problematic since the evaluation of the paper shows that the Mask-RCNN performs badly on NAO.
>
> The mask generator in mask-RCNN is label independent so the accuracy of the mask is less dependent on the object category, whether it is correct or not. However, we recognize that Mask-RCNN could perform worse on objects that it has not seen. We visually inspected several of the masks generated by Mask-RCNN on some natural adversarial objects and we found the predicted mask acceptable. In the future, we can further annotate the instance masks for NAO which will provide more accurate object masks for this experiment.
>
> > The paragraphs on (natural) adversarial examples are rather short and contain only few references.
>
> We expanded the section about natural adversarial examples and cited the paper listed.
>
> > The set of detection models evaluated on NAO includes EfficientDet D7, which was used in the generation of NAO. Low performance of EfficientDet D7 on NAO is to be expected, as the samples for NAO were selected specifically because EfficientDet D7 performed badly on them.
>
> We recognize that it is not surprising that EfficientDet-D7 performs badly on NAO. We include EfficientDet-D7 in our evaluation graph and table to provide a reference showing the reduction in accuracy of the models are similar across multiple types of object detection models.
>
> > The authors might want to take another look at the information given in the supplementary material and extend/correct it where appropriate.
>
> We thank the reviewer for spotting the errors and typos in the paper. We have fixed and clarified the documentation and construction about NAO in the appendix.

---

> > ### Comment · Reviewer_Kcpe · 2021-07-13
> > **Response to the authors**
> >
> > Thank you very much for the explanations and clarifications. I will update parts of the review.
> >
> > > The meaning of OOD in this metric is objects that do not belong to the MSCOCO label categories. For example, if the model predicts a shark as an airplane, we do not penalize the model for this prediction because it has not seen a shark before. However, we recognize that if the airplane is OOD (e.g. an old propeller plane instead of a jet plane), this metric does not exclude such OOD objects that are part of the MSCOCO label categories.
> >
> > I think there is still a problem because OpenImages is not exhaustively annotated. Therefore, there can still be false positive predictions on OOD objects (that do not belong to a MSCOCO label category) which are not excluded from the mAP-without-OOD, because there are no annotated bounding boxes for these objects in OpenImages.
> >
> > > The mask generator in mask-RCNN is label independent so the accuracy of the mask is less dependent on the object category, whether it is correct or not. However, we recognize that Mask-RCNN could perform worse on objects that it has not seen. We visually inspected several of the masks generated by Mask-RCNN on some natural adversarial objects and we found the predicted mask acceptable. In the future, we can further annotate the instance masks for NAO which will provide more accurate object masks for this experiment.
> >
> > Thank you for the explanation. For future work, one could also check whether the segmentation masks added in OpenImages V5 can be used for this. (Unfortunately, segmentation masks are not available for all objects annotated with bounding boxes.)
> >
> > > We recognize that it is not surprising that EfficientDet-D7 performs badly on NAO. We include EfficientDet-D7 in our evaluation graph and table to provide a reference showing the reduction in accuracy of the models are similar across multiple types of object detection models.
> >
> > I see the point in including EfficientDet-D7 in the evaluation as a reference number. For the interpretation of the results, the role of EfficientDet-D7 in the dataset creation is an important information. Maybe this could be indicated, e.g., in Figure 1? Otherwise, someone who does not read the paper in detail might miss this fact.

---

### Official Review · Reviewer_xJP8 · 2021-07-05
**The datasets are carefully collected, but the potential value for research is limited.**

**Rating:** 5
**Confidence:** 2
**Clarity:** The paper is well written and general…

**Strengths:**

+ The paper explains the source of data in detail, and the screening process is reasonable and prudent.
+ The authors evaluate 7 SOTA models on the proposed NAO dataset, reporting the mAP, mAR, and visualized failure modes.  The evaluation is multidimensional, and the statistical features of NAO is discussed comparing to MSCOCO and openImages.
+ The analysis and discussion about background cues and integrated gradients are sound and enlightening.

**Weaknesses:**

My main concern is that the proposed dataset itself does not go beyond the limitations of the OpenImages dataset. Specifically, the data set is acquired by evaluating the model trained on MSCOCO dataset.  The model is evaluated on the OpenImages dataset with more categories and data amount, to find out the failure data and form the NOA dataset. I think this dataset is more of a reflection of the distribution defects of MSCOCO dataset comparing to OpenImages data set.  Though labels of OpenImages are further corrected and modified, it seems this doesn't affect much.

On the other hand, the error tendency of the prediction model may change with the change of prediction framework and network structure in the future. Therefore, the effectiveness of fine-tuning or evaluating with DOA data sets may be gradually reduced with the improvement of SOTA, and may be effective only for a certain group of models.

Although the author has done detailed work in data analysis and processing, I think the dataset can provide limited value to the research community.

**Additional Feedback:**

Please refer to weaknesses.

**Correctness:**

The construction of the dataset is reasonable.  The claim in the paper is generally correct.

**Documentation:**

The documents are provided including data collection,  content list, usage and maintenance.

**Ethics:**

No ethics problem found for this dataset.

**Relation To Prior Work:**

There is sufficient discussion regarding related works.

**Summary And Contributions:**

The paper presents Natural Adversarial Objects (NAO) dataset, which is used for robustness evaluation for detection models trained on MSCOCO dataset.  To obtain the adversarial data, a model trained on MSCOCO dataset is evaluated in OpenImages that is on a larger scale and owns more categories.  A rigorous data filtering and correcting process is applied to find the hard false positives and egregiously misclassified cases. The authors study the distribution of OpenImage, MSCOCO, and NAO, and believe the dataset facilitates further research about model robustness and handling out-of-distribution inputs.

---

> ### Author Response · Authors · 2021-07-09
> **Response to the research value**
>
> We thank reviewer xJP8 for the review. Our responses to the review are as follows:
>
> > My main concern is that the proposed dataset itself does not go beyond the limitations of the OpenImages dataset.
>
> OpenImages is currently one of the largest open source dataset with bounding box annotations. While the dataset is still limited (e.g. lacking visual grounding, etc.), we believe sourcing from OpenImages would produce a diverse natural adversarial dataset. We recommend using NAO as an additional test set for models trained on MSCOCO dataset and evaluated the model robustness against edge cases and out of distribution samples.
>
> > On the other hand, the error tendency of the prediction model may change with the change of prediction framework and network structure in the future.
>
> It is possible that future models might have different performance characteristics on different models but we have evaluated seven SOTA models from the past few years and observed that the past models with different architectures all perform badly on MSCOCO dataset.
>
> > Although the author has done detailed work in data analysis and processing, I think the dataset can provide limited value to the research community.
>
> Majority of the adversarial data research focuses on the synthetically generated adversarial examples, not the naturally occurring ones. This dataset is one of very few to provide such a benchmark for object detection models. We believe our analysis into the reason why models are prone to natural adversarial objects can motivate future research in naturally occurring edge cases, and therefore making the model more robust especially in safety-critical applications.

---

### Decision · Program_Chairs · 2021-07-26

**Decision:**

Reject

**Comment:**

The paper proposes a challenging dataset to evaluate the robustness of object detection models under "natural adversarial objects".  Reviewers found the paper/dataset relevant and interesting, but overly narrow given the dataset and pre-training context.